Riccardo Di Sipio, University of Toronto

riccardo.disipioutoronto.ca

August 7, 2019

## Abstract

**A likelihood-based unfolding method based on Bayes' theorem is presented, with a particular emphasis on the application to differential cross-section measurements in high-energy particle interactions.**

## 1 Introduction

Unfolding is the procedure of correcting for distortions due to limited resolution of the measuring device [1]. In this paper, we present a likelihood-based approach to unfolding similar in nature to the Fully Bayesian Unfolding technique [2,3] called *Eikos* [1]. An iterative method to stabilize the initial trial function is employed, in a fashion similar to the method introduced in [4]. The method is implemented having in mind the application to differential cross-section measurements in high-energy particle interactions, where the data are the rates of observed

---

[1]From the ancient Greek word which is often translated as probability, plausibility, or likelihood.

collisions as a function of the independent variable of physical interest, such as the momentum of one of the particles produced in the interaction. It provides as output the most common kind of information expected in this kind of measurement: absolute and normalized unfolded distributions, the total cross-section and correlation matrices. Markov Chain Monte Carlo (MCMC) integration [5], marginalization and plotting are performed by relying on the Bayesian Analysis Toolkit (BAT) package [6]. The code is publicly available on the CERN GitLab repository https://gitlab.cern.ch/disipio/Eikos .

For this application, Bayes' theorem can be stated as follows:

$$P(\sigma, \theta) \propto \mathcal{L}(d|\sigma, \theta)\pi(\sigma)\pi(\theta), \tag{1}$$

where $\sigma$ represents the measured value of the cross-section, $\theta$ is a set of nuisance parameters (NPs) corresponding to sources of systematic uncertainties in the measurement, $\pi$ is the prior distribution representing our *a priori* belief about the cross-section ($\pi(\sigma)$) and the systematics ($\pi(\theta)$), *i.e.* our initial guess of the probability distribution of $\theta$, $\mathcal{L}$ is the likelihood of observing a certain number of data events $d$ given $\sigma$ and $\theta$, and finally $P$ is the posterior distribution, *i.e.* the probability of a combination of $\sigma$ and $\theta$ given the number of observed data events $d$. The likelihood of the data given a predicted spectrum $\mu$ (which in the most general case includes both signal events and expected background sources) and a set of systematic uncertainties $\theta$ is estimated by a counting experiment:

$$\mathcal{L}(d|\sigma, \theta) = \prod_{i}^{N_{reco}} Poiss(d_i|\mu_i(\sigma, \theta)) = \prod_{i}^{N_{reco}} \frac{\mu_i^{d_i}}{d_i!}e^{-\mu_i} \tag{2}$$

$$\mu_i(\sigma, \theta) = S_i + B_i = \sum_{j}^{N_{truth}} (U_{ij}(\theta)\sigma_j) + B_i \tag{3}$$

$$\pi(\theta) = \prod_{k}^{N_{syst}} N_k(\theta) = \prod_{k}^{N_{syst}} \frac{1}{\sqrt{2\pi}s_k}e^{-\frac{1}{2}\left(\frac{x-\theta}{s_k}\right)^2}. \tag{4}$$

The cross-section in each bin $j$ of the unfolded data histogram $\sigma_j \equiv \frac{d\sigma_j}{dX}$ is extracted from the mean and RMS of the corresponding posterior distribution. In principle, the number of bins used for the observed data (or "reco" level)$N_{reco}$ and the unfolded results (or "truth" level) $N_{truth}$ do not have to match allowing also for over- and under-constrained unfolding. The parameters $s_k$ represent a unit of standard deviation associated with each of the $N_{syst}$ sources of systematic uncertainty, which are usually estimated from the simulation.

## 2 Mathematical definition

### 2.1 Forward Folding

The operator $U_{ij}(\theta)$ transforms a cross-section in the $j$-th truth bin into an event yield in the $i$-th reco bin. This operator can be expressed as the product of three matrices:

$$U_{ij}(\theta) = A_i^{-1}M_{ij}E_j = \begin{pmatrix} \frac{1}{a_0} & & 0 \\ & \ddots & \\ 0 & & \frac{1}{a_N} \end{pmatrix} \begin{pmatrix} M_{00} & \cdots & M_{0N} \\ \vdots & \ddots & \vdots \\ M_{N0} & \cdots & M_{NN} \end{pmatrix} \begin{pmatrix} \epsilon_0 & & 0 \\ & \ddots & \\ 0 & & \epsilon_N \end{pmatrix}, \tag{5}$$

where $A$, $M$ and $E$ represent acceptance, migration and efficiency corrections, respectively. Typically, $M$ is presented as a matrix of (reco, truth) with rows that are normalized to unity, so that each entry is the probability that events produced in a given truth bin will be observed in the specified reco bin. The inverse of this matrix multiplication is commonly known as *unregularized matrix inversion unfolding*.

## 2.2 Corrections

A model is typically employed to estimate the relationship between truth-level and reco-level quantities, and the most common implementation of the experimental effects such as detector efficiency and resolution are done using a Monte Carlo event generator followed by a stochastic simulation of the effects of the detector. Given a theory model, the set of acceptance, migration and efficiency corrections can be estimated from the truth-level spectrum $\sigma_j$ (events passing the particle-level fiducial selection), the reco-level spectrum $S_i$ (events passing the detector-level selection), and the response matrix (events passing both selections). In particular, we have that:

$$E_j = \sum_i U_{ij}/\sigma_j \tag{6}$$

$$A_i = S_i/\sum_j U_{ij}, \tag{7}$$

where the two summations are equivalent to a projection of the migration matrix.

## 2.3 Systematics

There are various sources of systematic uncertainty arising from the models used to generate events and the effects of the detector, and these are captured by introducing additional nuisance parameters into the likelihood. In particular, the correction terms depend on these nuisance parameters, which can be classified in two main groups:

- modelling systematics associated with the event generation process have different truth-level spectra (and so reflect differences in model), hence all the corrections are unique to each alternative model;

- detector systematics are independent of any consideration concerning the same truth-level spectrum, hence all have the same efficiency;

The effect of the $k$-th systematic $\theta_k$ is parametrized as a Gaussian shift $\lambda_k \sim N(0,1)$, completely correlated across the bins:

$$\mu_i = S_i + B_i \tag{8}$$

$$S_i = S_i(\theta = 0) + \sum_{k \in syst} \lambda_k \Delta S_i(\theta_k) \tag{9}$$

$$B_i = B_i(\theta = 0) + \sum_{k \in syst} \lambda_k \Delta B_i(\theta_k). \tag{10}$$

When $\lambda_k = \pm 1$, the shift corresponds to a 1 standard-deviation ($\pm 1\sigma$) variation ($\Delta S$ and $\Delta B$) with respect to the nominal. Asymmetric systematic uncertainties are described by a two-sided Gaussian, *i.e.* for $\lambda_k > 0$, $\sigma = \sigma^+$ and for $\lambda_k < 0$, $\sigma = \sigma^-$. In general, some assumptions

have to be made in order to estimate the shifts $\Delta S(\theta)$ and $\Delta B(\theta)$, and these either introduce additional sources of uncertainty or become limitations on the unfolding results.

# 3    Details of the implementation

The *Eikos* program is based on BAT, which makes use of a Markov Chain Monte Carlo (MCMC) calculation to sample the parameter space and maximize the posterior joint probability distribution. A number of parameters have to be defined in order to calculate the differential cross-sections with the MCMC, and at the same time to take into account the systematic shifts. Finally, it is possible to define a number of spectator variables that are defined as functions of the MCMC parameters.

In *Eikos* , the MCMC parameters are $N = N_{bins}$ relative shifts with respect to a trial distribution, which is usually equivalent to the nominal prediction obtained by running the Monte Carlo event generator. The numerical value of each of these parameters is usually of order unity. An additional $K$ parameters are added to incorporate the effects of the systematic uncertainties. Finally, $2N + 1$ additional observables are defined to calculate the absolute differential, normalized differential and inclusive cross-sections. These observables are calculated at each iteration of the MCMC, based on the numerical value of the MCMC parameters.

For each point, the posterior distribution $P$ is calculated as the product of the prior times the likelihood. For practical and numerical stability reasons, the current implementation requires the definition of the logarithm of the prior and of the likelihood, *i.e.* :

$$\log P = \log \mathcal{L} + \log \pi(\sigma) + \log \pi(\theta). \tag{11}$$

## 3.1    Prior distributions

For unregularized unfolding, a flat prior for the cross-section is used, *i.e.* $\pi(\sigma)$ is a constant function between $[0, 2]$, where 1 corresponds to the center of a reasonably large range around the value $\sigma_0$ of the cross-section of the trial model used as starting distribution. In an extreme case, the trial distribution can be a constant across the entire spectrum, but usually it corresponds to the nominal theoretical model obtained by running the Monte Carlo event generator. The range of the prior in each bin may have to be adjusted in some particular cases.

For regularized unfolding, a multinormal prior is used, *i.e.* $\pi(\sigma)$ is the sum of positive-definite Gaussian distributions $N(0, 0.3)$. It is worth noting the a 30% uncertainty is usually quite large, but it is helpful in this example to illustrate how the effects of the systematic uncertainties are incorporated by *Eikos* into the results. The actual values would normally be estimated as part of the measurement process.

Finally, each nuisance parameter $\theta$ representing one of the systematic uncertainties has a corresponding prior $\pi(\theta)$ equal to a Gaussian distribution with mean 0 and standard deviation 1. For this purpose, the nuisance parameters have been scaled in such a way that a standard deviation of unity reflects the appropriate uncertainty.

A feature of the *Eikos* unfolding is the possibility to iterate multiple times the integration of the cross-section posterior in order to find a trial distribution that is in better agreement with the observed data before executing the full-fledged marginalization including the systematics,

*i.e.* :

$$P_1(\sigma, 0|d) \quad \propto \quad \mathcal{L}(d|\sigma, 0)\pi_0(\sigma) \tag{12}$$
$$P_2(\sigma, 0|d) \quad \propto \quad \mathcal{L}(d|\sigma, 0)P_1 \tag{13}$$
$$\ldots$$

An iterative method was first introduced in a paper by D'Agostini [4], where Bayes' theorem is used as part of a rule to update initial estimates for the cross sections, but not the posterior distributions (otherwise data would be used multiple times to artificially reduce the total uncertainty). In *Eikos* , in order to find an initial trial cross-section distribution, an initial flat prior $\pi_0(\sigma)$ is used in the first iteration, which is equivalent to an unregularized matrix inversion. Systematic uncertainties are not considered in the trial-finding step. In subsequent iterations, the posterior obtained in the previous step is set as a the new trial distribution, and the multinormal prior described above is used to converge towards a stable trial distribution. At the end of these iterations, the final distribution is stored so that it can be used as the trial distribution in the following steps, where the posterior is integrated by taking into account systematic uncertainties.

## 3.2 Likelihood

At each iteration of the MCMC, the trial differential cross-section is represented by a tuple of $N$ random parameters $\hat{\sigma} = (\sigma_1, \ldots, \sigma_N)$. The reco-level central prediction ($\theta = 0$) is estimated using the forward folding:

$$\hat{S}_i(0) = A_i^{-1}(0)M_{ij}(0)E_j(0)\hat{\sigma}_j. \tag{14}$$

At this point, the reco-level prediction for the signal component is given by the following linear approximation:

$$\hat{S}_i = \hat{S}_i(0) + \sum_{k \in syst} \lambda_k \Delta S_i(\theta_k). \tag{15}$$

The effect of a systematic uncertainty $\Delta S_i$ is estimated before the likelihood minimization depending on the type of uncertainty:

- For a detector-level systematic uncertainty, the shift is calculated from the difference of predicted events at reco level with respect to the nominal sample:

$$\Delta S_i = S_i(\theta_k) - S_i(0). \tag{16}$$

- For modelling systematic uncertainty, the trial differential cross-section spectrum $\hat{\sigma}$ is forward-folded using corrections taken from the alternative model, and then compared to the nominal prediction at reco-level, with the latter kept fixed during the process:

$$\Delta S_i = U_{ij}(\theta_k)\hat{\sigma}_j - S_i(0). \tag{17}$$

Finally, it is possible to add a regularization term $\rho$ to favour smooth differential cross-sections that have certain characteristics such as smoothness. This is achieved by minimizing

the Tikhonov curvature as in [2] (second derivative), also taking into account the bin width:

$$\log \mathcal{L}' = \log \mathcal{L} - \alpha\rho \tag{18}$$

$$\rho = \sum_{j=2}^{N-1} \left| \frac{\Delta_{j+1,j} - \Delta_{j,j-1}}{\Delta_{j+1,j} + \Delta_{j,j-1}} \right| \tag{19}$$

$$\Delta_{m,n} = \frac{\frac{1}{w_m}\frac{d\sigma_m}{dX} - \frac{1}{w_n}\frac{d\sigma_n}{dX}}{x_m^0 - x_n^0}, \tag{20}$$

where $w_i$, $x_i^0$ and $\frac{d\sigma_m}{dX}$ are the width, center and differential cross-section of the $i$-th bin, respectively. Additionally, $\alpha$ is a parameter to be tuned that controls the strength of the regularization. It is suggested that $\alpha$ should take upon a numerical value close to that of the likelihood and prior functions. For example, a simple function of the number of bins $N$ and the number of systematics $K$ such as $(N + K)/N$ satisfies this requirement.

## 4 A practical application

A working example of the *Eikos* unfolding is provided by the use of a simulated experiment. The aim is to obtain a differential cross-section by unfolding to truth-level a reco-level distribution of a model observable $x$. The pseudo-data $D$ are sampled from a *p.d.f.* composed by the sum of a signal ($S$) plus a single background ($B$). A Gamma distribution is used to generate the signal distribution, while an exponential distribution is used to model the background:

$$D(x) = S(x) + B(X) \tag{21}$$

$$S(x) \sim \frac{1}{\Gamma(k)} \frac{x^{k-1}e^{-x/\theta}}{\theta^k} \tag{22}$$

$$B(x) \sim Ae^{-Bx}. \tag{23}$$

A total of 30,000 pseudo-data events have been generated with unit weight.

The efficiency, migrations and acceptance corrections and the expected background are obtained from statistically independent samples generated using the same model that is used for the pseudo-data. In particular, the same numerical values of the model parameters ($k = 2.5$, $\theta = 10$, $A = 50$, $B = 25$) are used. The expected signal events are generated with much higher statistics compared to the pseudo-data (1 million events) to reduce the statistical uncertainty and are thus scaled with a weight equal to the integrated luminosity ratio $w = L(data)/L(signal)$, where $L$ reflects the total integrated flux of collisions for both the pseudo-data and the modelled signal process. The number of pseudo-data and signal events are large enough to populate with sufficient statistics all the bins of the reco-level distributions and the migration matrix. These distributions are binned into a histogram with bins of variable width, as is often the case in recent analyses [7,8]. This is illustrated in Fig. 1.

Efficiency and acceptance corrections are simulated by a constant function with value equal to 0.30 and 0.80, respectively, as shown in Fig. 2. Migration matrices are emulated by smearing the generated value $x$ by a Gaussian with mean zero and $\sigma = \sigma_0 + c * x$. The resulting migration matrix, whose elements on each row add up to unity, is shown in Fig. 3.

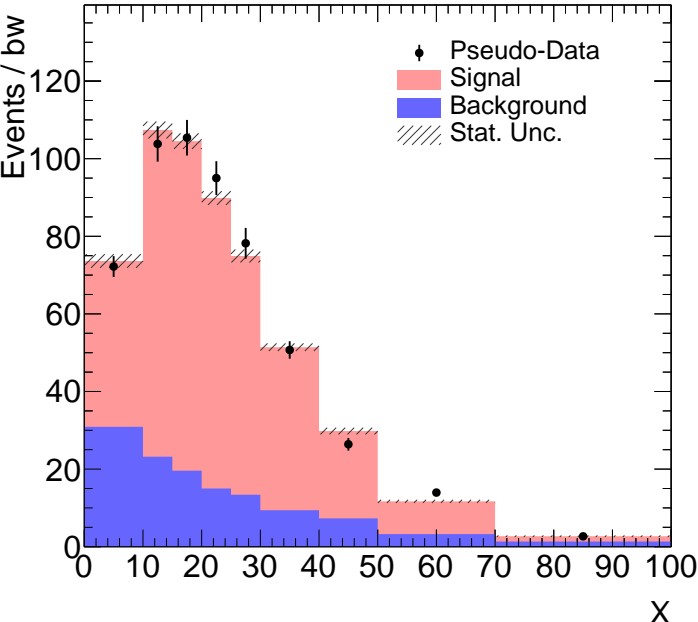

Figure 1: Detector-level distribution of the pseudo-data in the model observable $x$. The black dots represent the pseudo-data. The filled areas represent the stacked histograms of the expected signal and background.

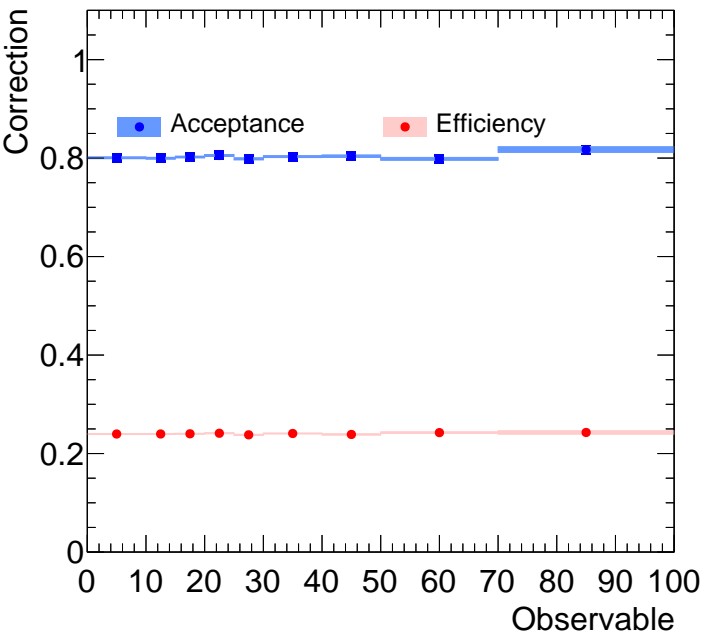

Figure 2: Efficiency and acceptance corrections. The shaded areas represent the statistical uncertainty.

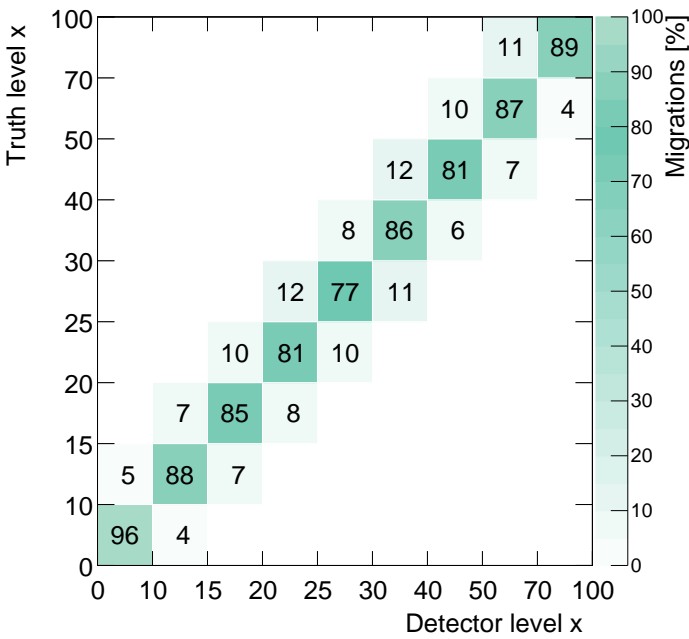

Figure 3: The similated migration matrix. The elements on each row add up to unity.

The consistency of any unfolding method is checked by two type of tests, closure and stress, which results are presented in Fig. 4.

A *closure test* is performed to make sure that any distortion possibly introduced by the unfolding procedure is not large and no bias is introduced. Ideally, one should recover the original distribution. However, limited statistics and the choice of regularization method usually have an effect on the unfolding. In this kind of test, a reco-level distribution generated with a given model $A$ is unfolded using corrections derived from the same model, and then compared to a truth-level distribution obtained using the same model.

In a second step, the *stress test* is carried out to determine if any bias is introduced in the unfolding results by the assumption of the model chosen for the corrections. A certain model $B$ is used to generate the truth- and reco-level distributions, while model $A$ (usually corresponding to the nominal) is used to perform the unfolding. The unfolded differential cross-section is then compared to a truth-level distribution obtained using model $B$. Also in this case, the ratio between the truth and unfolded distributions should be as close as possible to unity in each bin.

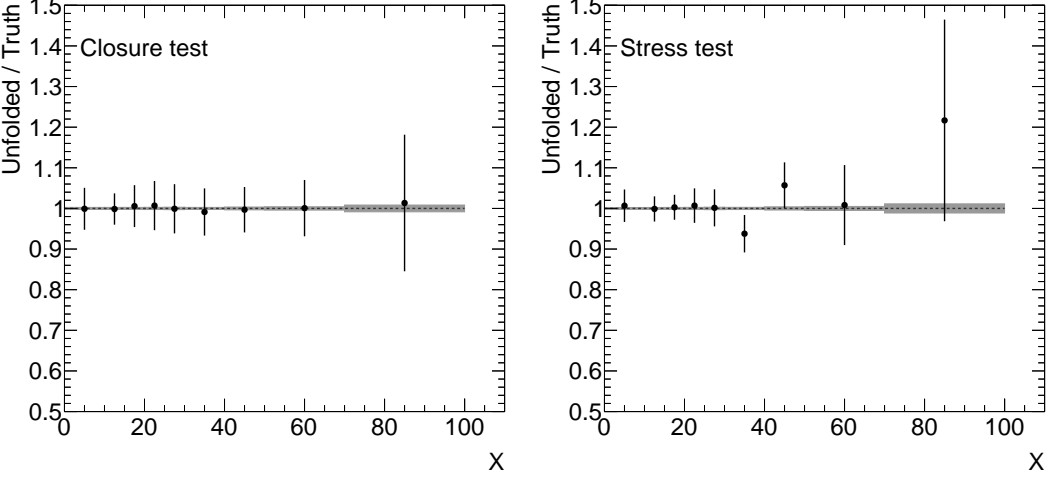

Figure 4: Closure (left) and stress (right) tests. Efficiency, migrations and acceptance corrections are estimated using the nominal model. The model with alternative value of the $\theta$ parameter is used as pseudo-data for the stress test. The black dots represent the ratio between the unfolded and the truth distributions. The hatched area represents the statistical uncertainty of the truth-level distribution.

Systematic uncertainties are emulated as follows:

- multiplicative systematics: the value of the reco-level $x_{reco}$ is scaled by a factor $x' = ax$;

- additive systematics: the value of the reco-level $x_{reco}$ is shifted by $x' = x + b$;

- weight systematics: the event weight is scaled by a factor $w' = aw$; and

- modelling systematics: the $(k, \theta)$ parameters of the Gamma distributions are shifted.

Posterior distributions of the parameters corresponding to the differential cross-sections are typically constrained by data with respect to the corresponding prior. In principle, the posterior distributions can also be pulled, *i.e.* the means are shifted with respect to the central value of the prior.

These effects can be seen in Fig. 5, where the posteriors (blue histograms) are always narrower than the priors (red histograms). In some cases, a moderate shift in the position of the mean is visible. By adding the value of the cross-sections in each bin of the differential distribution, the inclusive fiducial cross-section is obtained. Fig. 7 shows the update of the posterior distribution corresponding to the inclusive fiducial cross-section. It is evident that data prefer a larger value compared to the nominal prediction. The uncertainty is also reduced by the fitting procedure.

Posterior distributions of the parameters corresponding to the systematic uncertainties are not expected to be strongly pulled or constrained, otherwise indicating incorrect assumptions about their behavior. Fig. 8 shows a few cases where the posterior is unchanged, pulled or both. Fig. 9 summarizes the same information for all sources of systematic uncertainties.

The result of the application of the *Eikos* unfolding method to the input data is shown in Fig. 10. A comparison of the results from the *Eikos* method, the Matrix Inversion and the Iterative Bayesian (IB) unfolding is shown in Fig. 11. Very good agreement is observed between these three approaches. However, while in the IB approach each systematic has to be unfolded individually, hence loosing information about correlations, *Eikos* preserves such correlations and those among differential cross-sections bins and systematic uncertainties.

The agreement between theoretical models and the measured differential cross-section is often estimated by calculating the $\chi^2$ and the corresponding $p$-values. To do so, it is necessary to estimate the covariance, or equivalently the strength of the correlations, between the bins of the distribution. Once the correlations are known, the covariance $C_{ij}$ between bin $i$ and bin $j$ is simply given by:

$$C_{ij} = c_{ij}\sigma_i\sigma_j, \tag{24}$$

where $c_{ij}$ is the correlation factor, $\sigma_i$ and $\sigma_j$ are the uncertainties in bin $i$ and $j$, respectively. The uncertainty in a given bin $i$ is thus equivalent to $\sqrt{C_{ii}} = \sqrt{\sigma_i^2} = \sigma_i$.

In the *Eikos* framework, this is achieved by calculating the correlation factor for each two-dimensional marginalized distribution, corresponding to pairs of bins of the absolute or normalized differential cross-section, as shown in Fig. 12. The effect of systematic uncertainties is taken into account implicitly by the very nature of the likelihood-based approach.

Fig. 13 shows the complete correlation matrices for the absolute and normalized differential cross-sections. The matrices obtained with a simple toy MC show, at least qualitatively, some general properties such as the appearance of anti-correlations in the normalized spectra, which is an effect of the normalization constraint. Remarkably, the marginalization can be applied to an arbitrary pair of variables that appear in the joint posterior probability distribution function. Most notably, a strength of the method is the ability to measure the correlations among systematics, which are often assumed to be uncorrelated. This was investigated by adding two systematics that are linearly anti-correlated. As can be seen in 14, the 2D marginalized posterior distribution is able to recover such correlation to a good degree. Application to real-life cases beyond the toy MC presented in this paper may give some useful insight *e.g.* into the interplay between the modelling and detector-level systematic uncertainty sources.

As a final check, non-square migration matrices can over- or under-constrain the unfolding

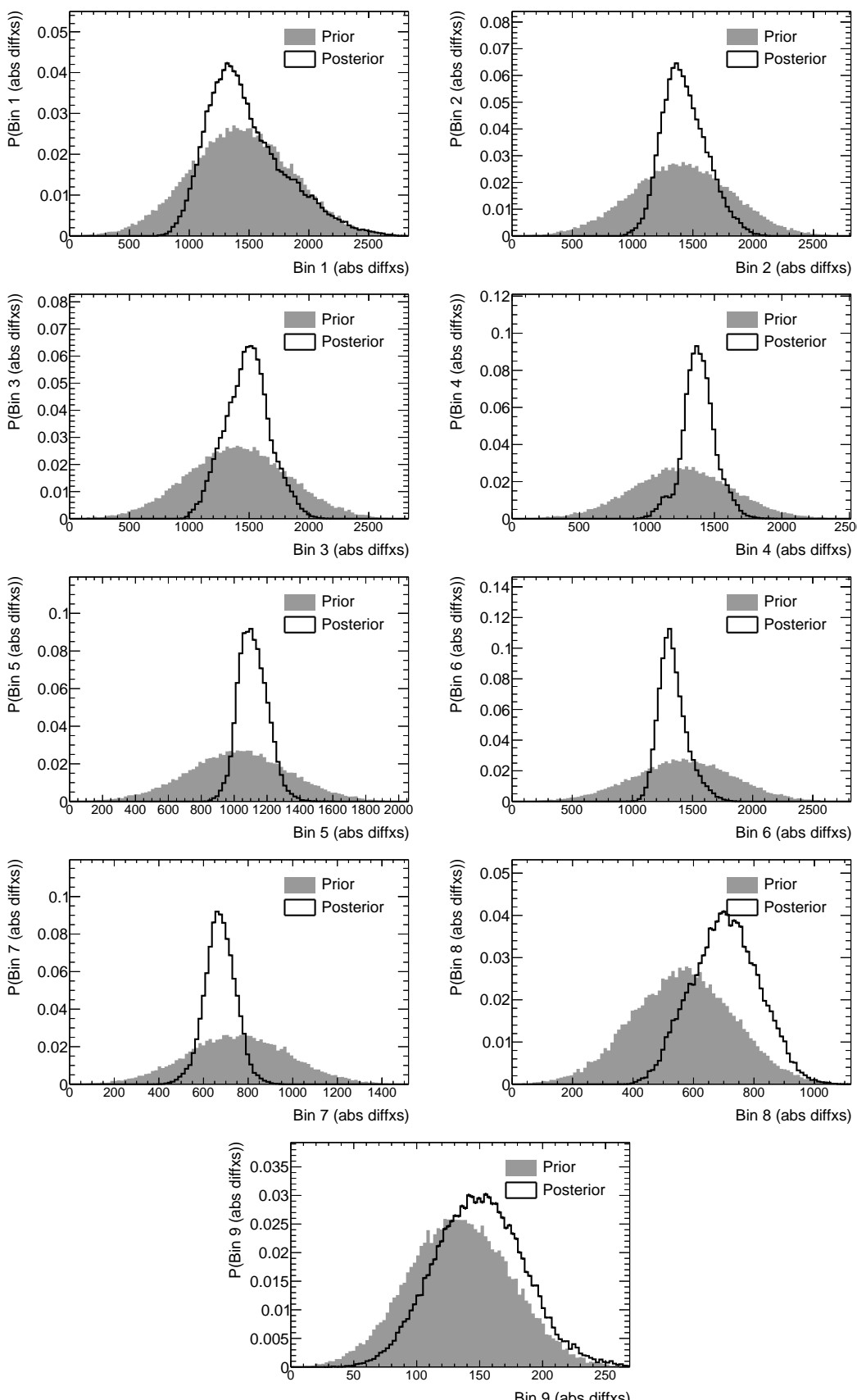

Figure 5: Update of posterior distributions of the parameters corresponding to the absolute differential cross-section.

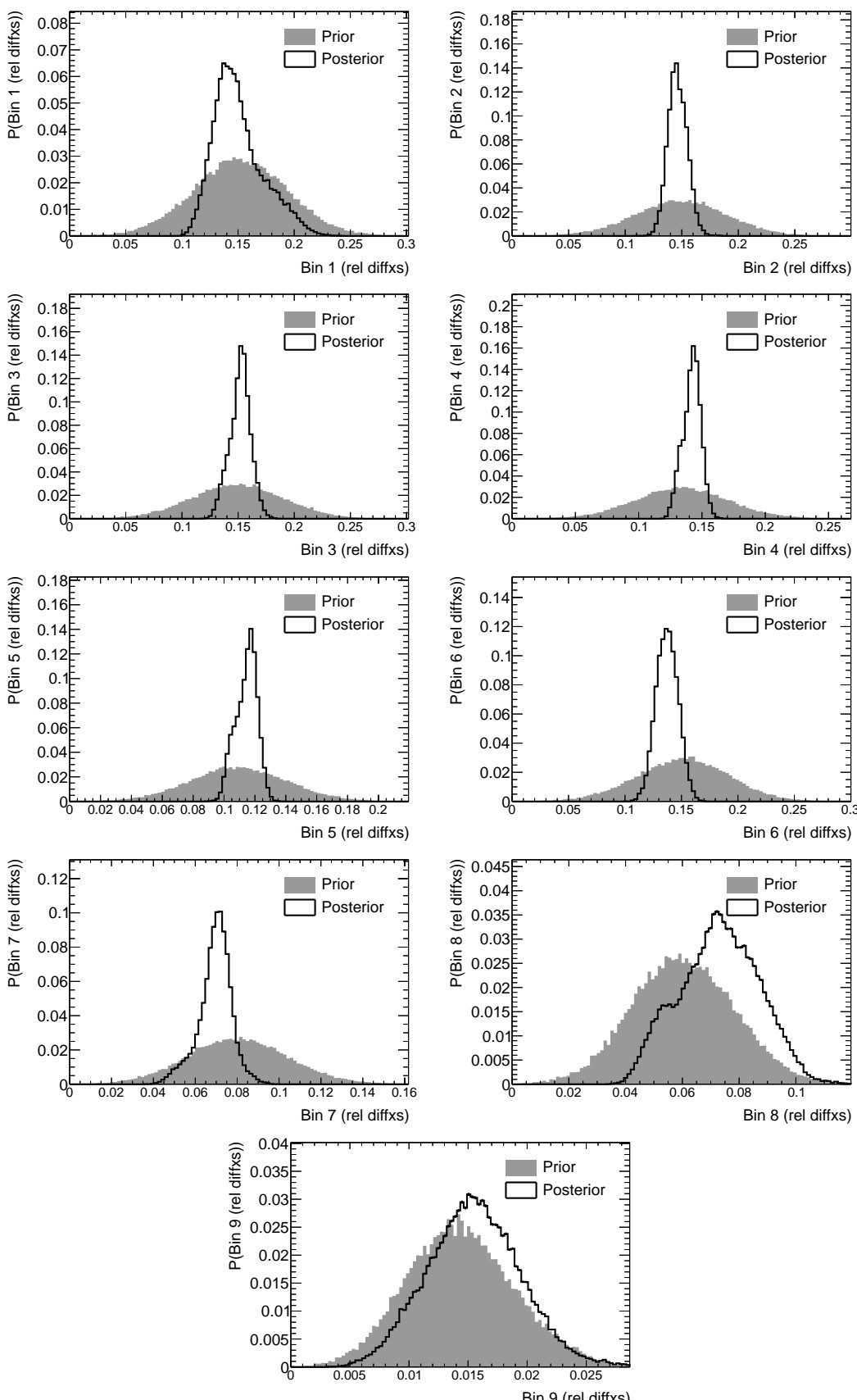

Figure 6: Update of posterior distributions of the parameters corresponding to the normalized differential cross-section.

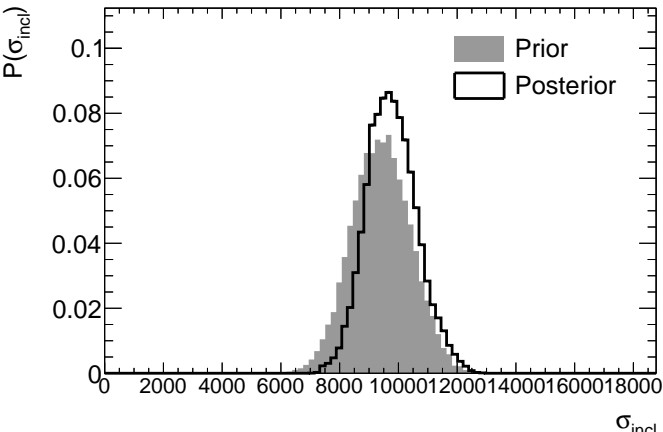

Figure 7: Update of posterior distributions of the parameters corresponding to the inclusive fiducial cross-section.

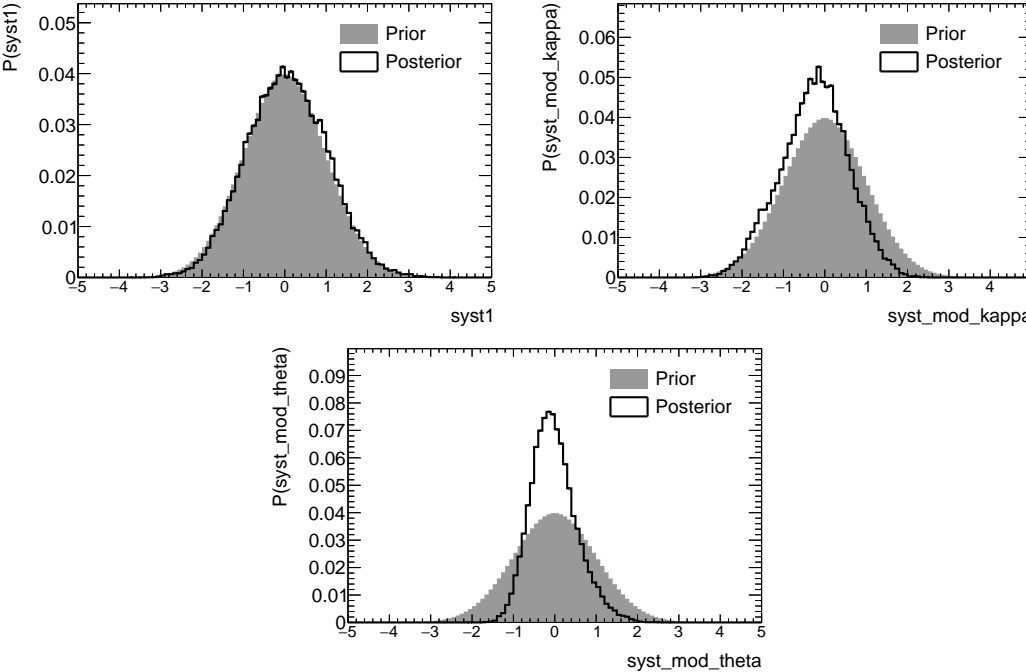

Figure 8: Update of posterior distributions of the parameters corresponding to the source of systematic uncertainty.

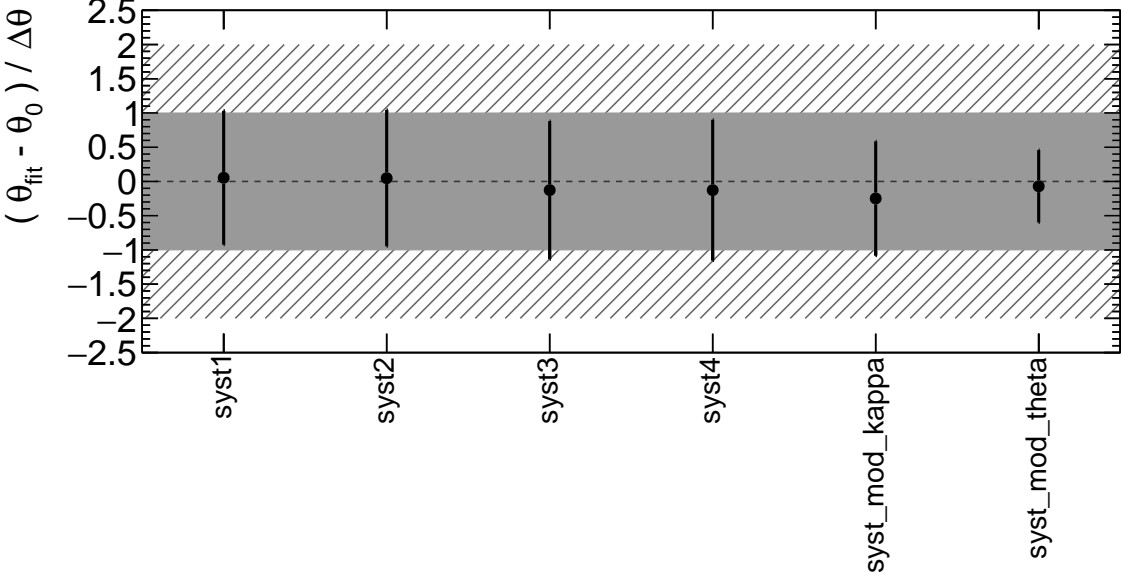

Figure 9: Summary of the update of posterior distributions of the parameters corresponding to sources of systematic uncertainties affecting the absolute differential cross-section.

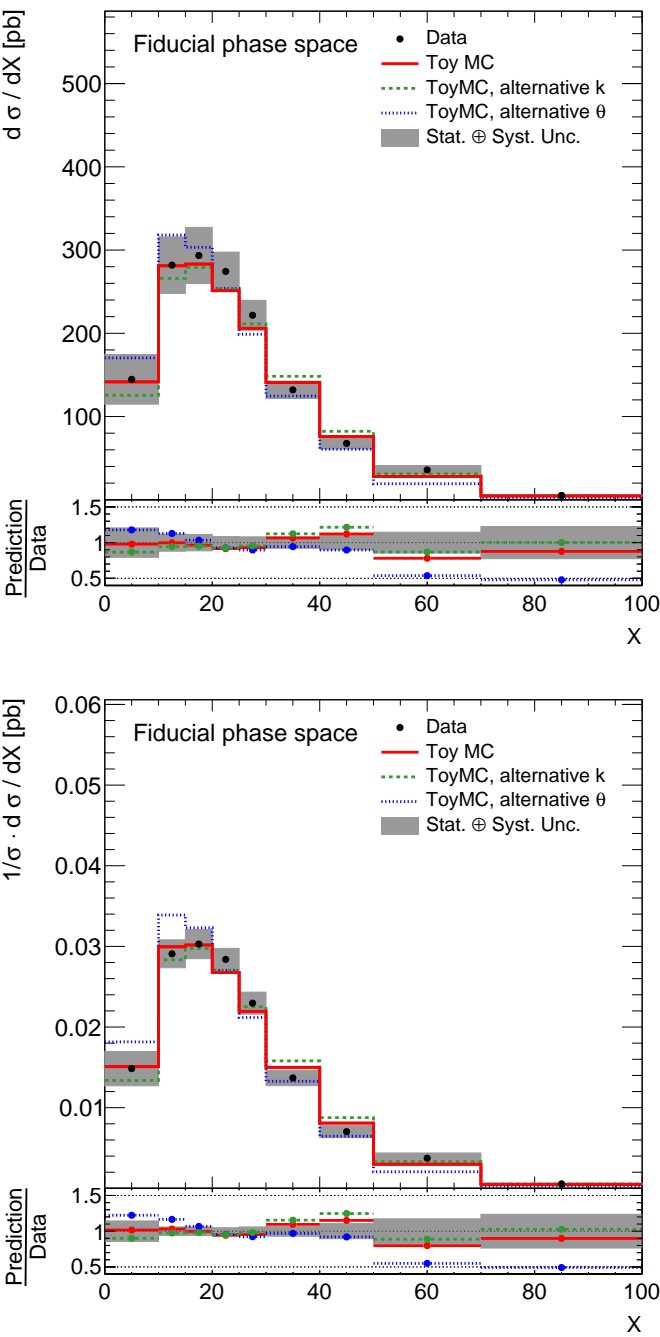

Figure 10: Absolute (top) and normalized (bottom) differential cross-sections. Alternative models with different values of $k$ and $\theta$ are also compared against the nominal.

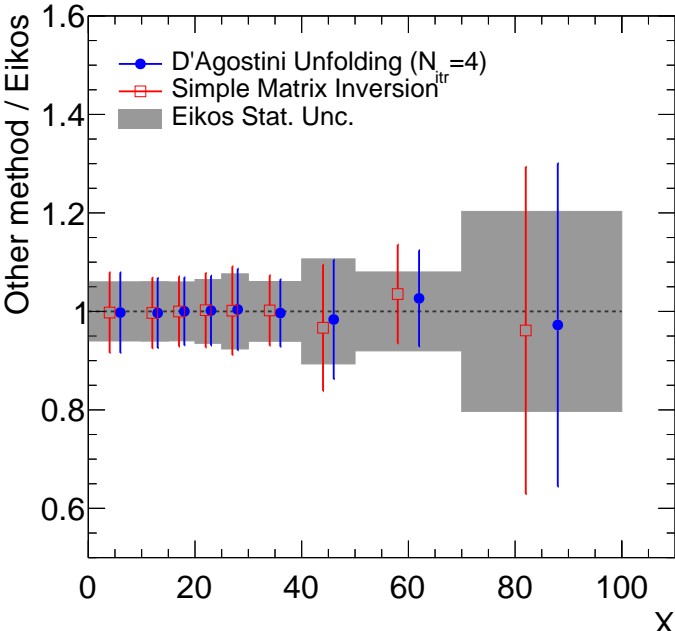

Figure 11: Comparison of the differential cross-section $d\sigma/dx$ unfolded with *Eikos* with respect to simple Matrix Inversion and D'Agostini iterative method with $N_{itr}$=4. The dots represent the ratio between the alternative methods and the *Eikos* unfolding. The hatched area represents the statistical uncertainty of the *Eikos* unfolding.

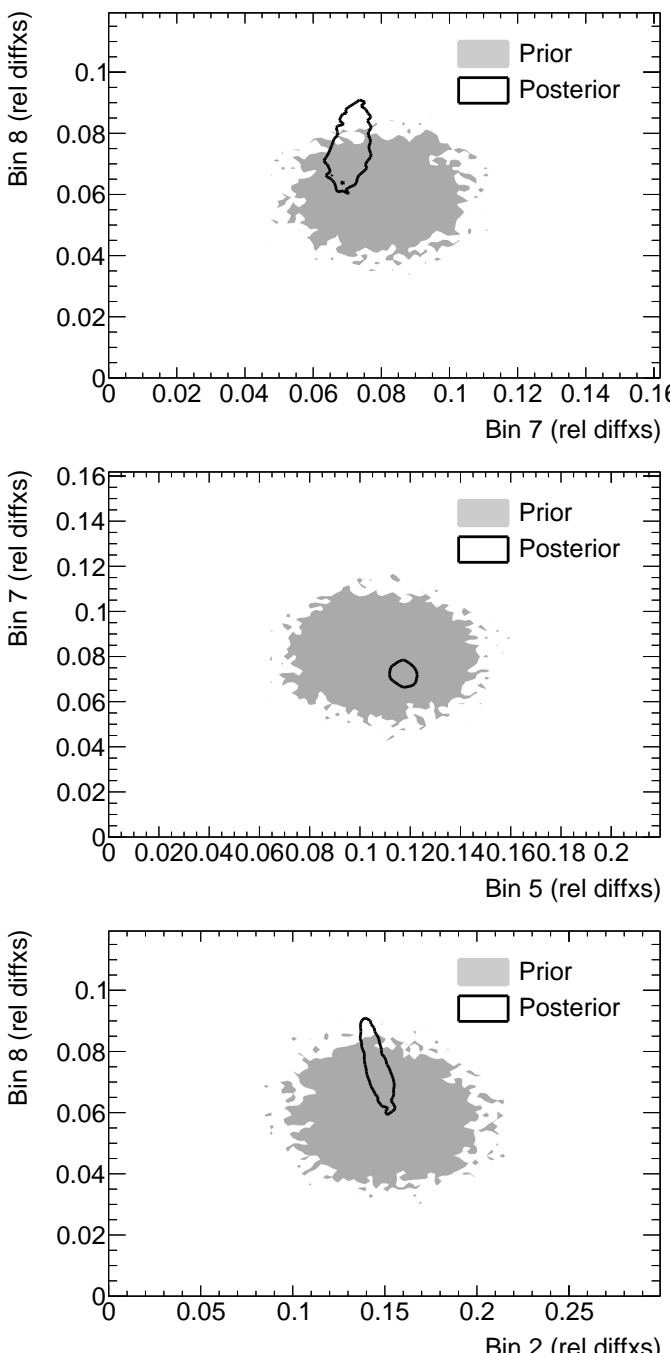

Figure 12: Examples of two-dimensional marginalized posterior distributions showing positive correlation (top), no correlation (centre) and anti–correlation (bottom) among bins of the normalized differential cross-section.

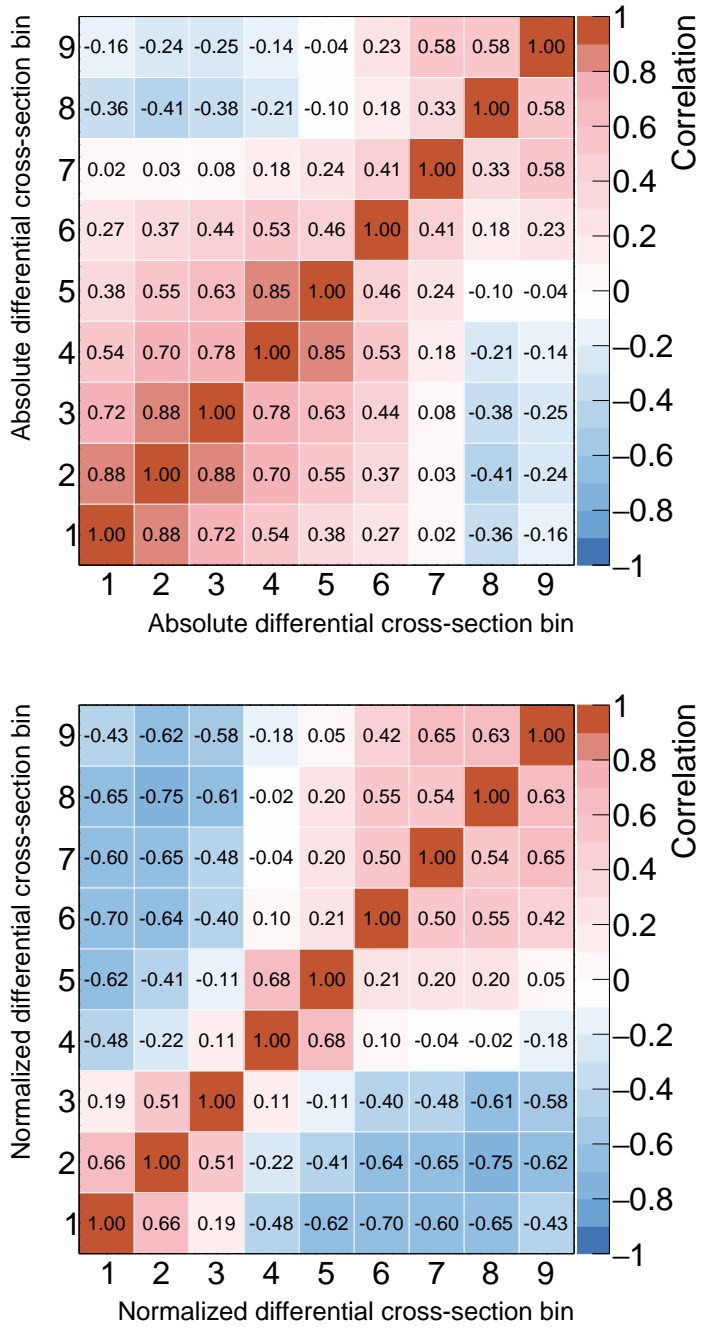

Figure 13: Correlation matrices between bins of the absolute (top) and normalized (bottom) differential cross-sections. The appearance of anti-correlations in the normalized differential cross-section is an effect due to the normalization constraint.

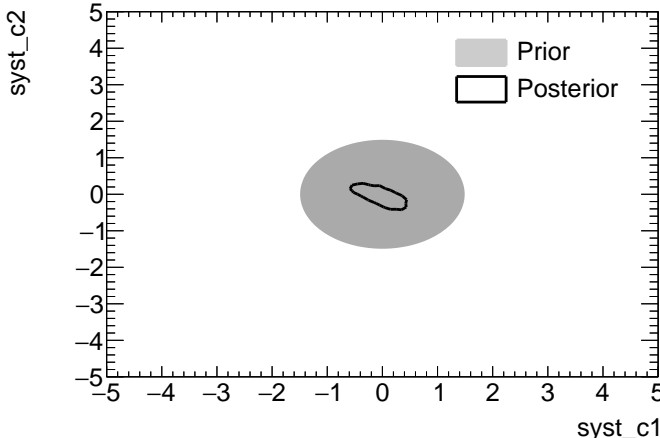

Figure 14: Example of two-dimensional marginalized posterior distribution showing anti-correlation between two sources of systematic uncertainty.

results, allowing for more or less stringent tests of specific theoretical predictions. In fact, in many cases the number of bins at reco level are limited by the data statistics, a problem that may not apply at truth level, where the number of simulated events is typically at least one order of magnitude larger. While a non-square matrix cannot be directly inverted, likelihood-based methods based on forward-folding do not suffer from this limitation. In this test, the nominal model was unfolded with nominal corrections, using three different binnings at reco level: underconstrained ($N_{reco} < N_{truth}$), overconstrained ($N_{reco} > N_{truth}$) and iso-constrained ($N_{reco} = N_{truth}$). Migration matrices for this three cases are shown in Fig. 15. Only the statistical uncertainty is considered. As shown in Fig. 16, the overconstrained differential cross-section is in very good agreement with the standard (isoconstrained) distribution. However, using fewer bins at reco level leads to larger uncertainties after unfolding. A fair agreement with the isoconstrained distribution is obtained by using a narrower Gaussian prior with $\sigma = 0.1$.

## 5   Conclusions and further developments

A likelihood-based approach to unfolding similar in nature to the Fully Bayesian Unfolding is presented. The implementation is aware of the peculiarities of differential cross-section measurement in particle physics applications. The posterior maximization procedure, carried out by the means of a Markov Chain Monte Carlo, gives the possibility to preserve the correlations among bins of the differential cross-section and the systematic uncertainties. As a case study, the method is applied to a toy Monte Carlo resembling a differential cross-section analysis. The results show that data can constrain the systematic uncertainties. Modelling systematics can be also pulled.

Future developments are likely to improve the current status of the *Eikos* approach. An open problem is the estimation of the separate effect of each source of systematic uncertainty. A typical approximate approach is to marginalize the posterior with all parameters left floating, and then run again $N_{syst}$ times by fixing $N_{syst} - 1$ parameters to their most probable

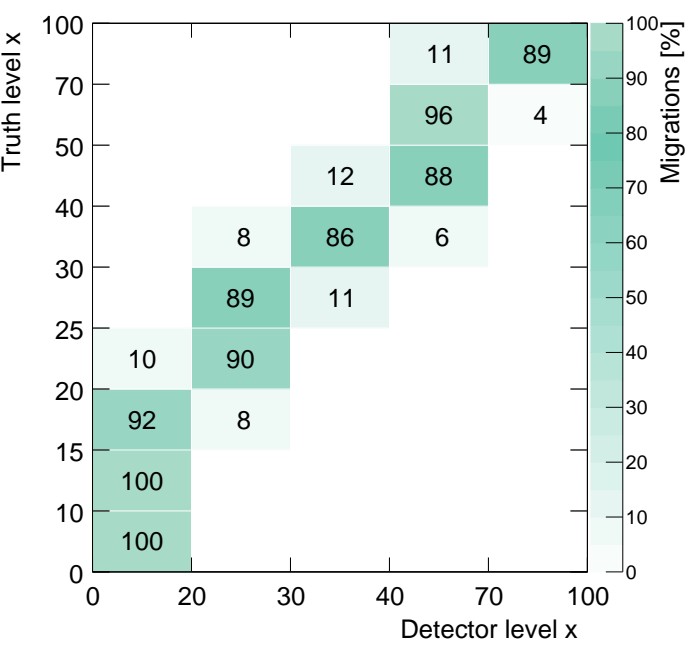

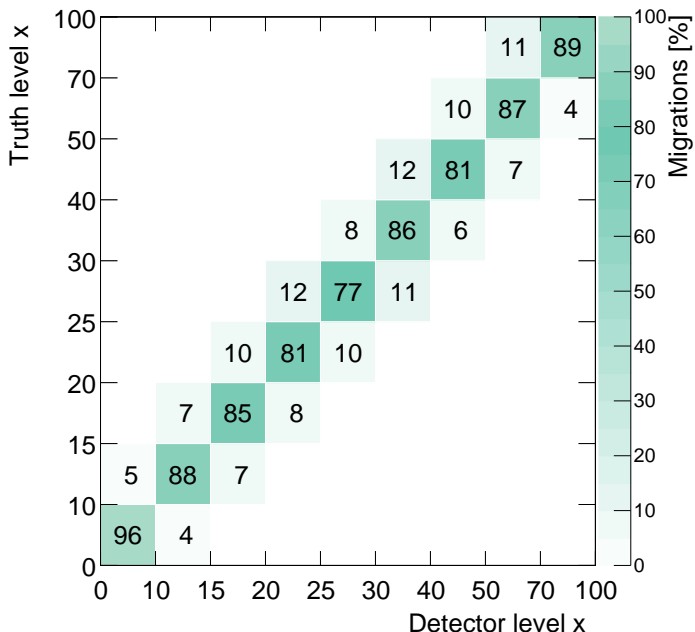

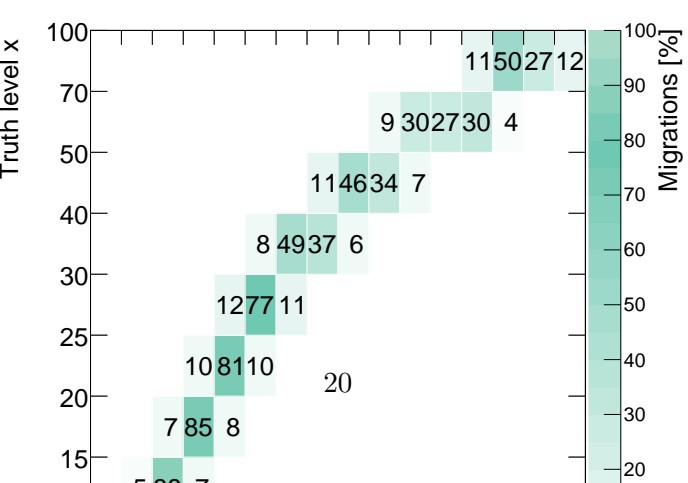

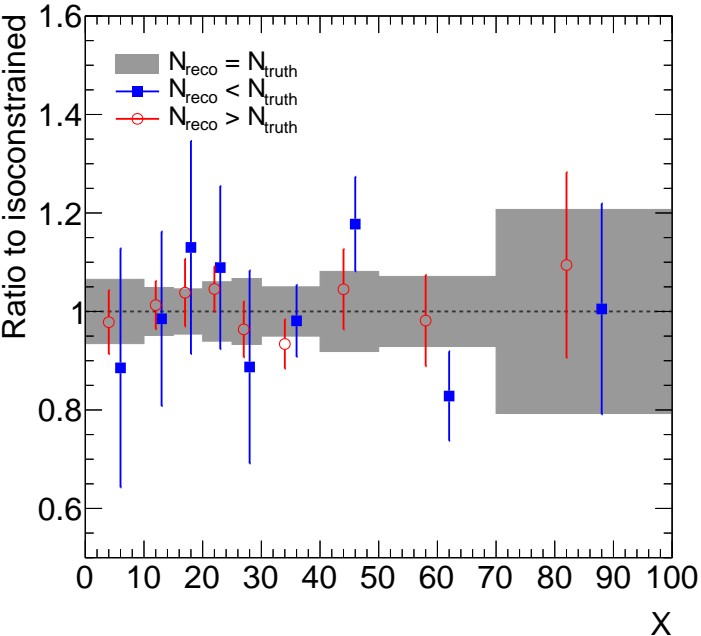

Figure 16: Comparison of the differential cross-section $d\sigma/dx$ unfolded with nominal corrections, using different binning at reco level. The hatched area represents the statistical uncertainty of the isoconstrained unfolded distribution.

value. However, is not expected to find exactly the same value of the total uncertainty for each run, as each minimization is carried out by the means of a stochastic algorithm (MCMC).

Finally, the Bayes' theorem provides a straightforward way to combine measurements performed in independent data samples. It is suggested that by the means of the combination, correlated uncertainties can be reduced in the minimization. A similar argument suggests that multiple variables can be unfolded at the same time, by adding more Poisson counting terms in the likelihood. This approach would provide a coherent estimation of a grand covariance matrix which can be used in turn, for example, in global fits of such theoretical quantities as parton distribution functions.

# Acknowledgments

I would like to thank Lorenzo Bellagamba (INFN Bologna), Pekka Sinervo (University of Toronto) and Francesco Spanò (Royal Holloway) for the useful discussions. I acknowledge the support of the Natural Sciences and Engineering Research Council of Canada (NSERC).

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
