# Peer review of "Eikos: a Bayesian unfolding method for differential cross-section measurements"

_SciPost Physics_

## Round 3 · Referee Report · Anonymous · 2019-9-18

Strengths

1. The paper is very relevant in the era of precision measurements at the LHC. Unfolding provides a solution to the need to preserve data for future interpretation, and this paper presents a method which aligns well with the procedures already adopted at the LHC.
2. The paper directs the reader to public routines which will help standardise the proposed unfolding strategy and allow for wider adoption of the methodology.
3. The use of a toy example is very helpful to see how the method works and provides a vehicle to investigate additional stress tests of the method.

Weaknesses

1. In a few instances, the paper refers to drawbacks of existing methods which are unfounded (these are detailed in the full comments)
2. In some areas, the paper is not clear on how the routine works so some clarity and definitions of some quantities in certain places would be helpful.
3. While the toy example is helpful in demonstrating the method, the paper could have been stronger by using a more realistic example or examples which demonstrate the limitations of the method.

Report

In general, the paper clearly describes an interesting modification to traditional unfolding methodologies, making direct use of experimental likelihoods. The method is supported with public routines, which allows readers to explore the use of the method in current measurements - for example at the LHC. Although the method is novel, the use of the likelihood for unfolding differential measurements has been used before (for example in https://doi.org/10.1016/j.physletb.2019.03.059), although estimation of the cross-section is performed via maximum likelihoods as opposed to MCMC integration. A reference to such uses would be appropriate.

One criticism is the suggestion that the method reported here allows for preservation of correlations between systematic uncertainties in the results where other methods (such as iterative Bayes), do not. However one could imagine unfolding (via IB) after varying each independent source of uncertainty and keeping track of the sign of the change in the resulting spectrum. This would allow one to maintain the correlation information. While its not as straightforward as the proposed method, it is still possible to do this so this statement seems unjustified.

Finally, the strength of the paper would be increased through more detailed exploration of the example model. For example, testing examples in which larger migrations between bins is present would showcase the use/need for regularisation and the robustness of the method to its use. Further discussion of compatibility between theoretical models and unfolded distributions with this method would be welcome (although potentially beyond the scope of this paper).

Requested changes

1. Section 1, P2, Para 2 : The last sentence before eqn 2 reads as though the likelihood is estimated, while it should read that the likelihood is *defined* by the following eqns 2-4.

2. Section 2.3 , P3, para 2: referring to a Gaussian shift seems confusing at this point. Later, the paper describes a prior for $\theta$ which is enough to define its distribution. I would reccomend to remove the term "Gaussian shift"

3. Section 3.1 : While the focus of this paper is not on Bayesian methods, it would be appropriate to comment on choices for the priors of the cross-sections in the context of differential measurements. Some motivation for the choices of these priors and potential alternatives would be welcome here.

4. Section 3.1, P4, para 2: The sum of Gaussians should be a product I think (since this refers to P, not Log(P)). Please clarify.

5. Section 3.2, P5, para 2: A "minimization" is referred to here, but as far as I could tell, no minimization has been described up until this point in the paper. Is this in error?

6. Section 4: The example given here seems to have very little migration between bins (judging from figure 3). Typically such cases do not require much regularisation. However, since regularisation is discussed, a mention of how the performance of the method behaves with larger migrations would be welcome.

7. Section 4, P10, para 3: Referring to parameters being pulled or constrained or "changed" is not clear in this context. The paper is presumably referring to the posterior distributions of the parameters. Instead, a definition of the "value" of the parameter before and after (thereby allowing for a definition of change), perhaps as the mode of the posterior, should be introduced here.

8. Figure 9: While an experimental particle physicist might be used to what $\Delta\theta$ and $\theta_{fit}$ means, the non-expert would be left to guess here. A definition would be welcome. Furthermore, the usual context of these quantities is under maximum likelihood fits. However, given that the method produces posterior distributions, these quantities are not obvious (eg is $\Delta\theta$ some quantile-range of the posterior distribution?).

9. Section 4: P10, para 5: It is good to see a discussion of how an unfolded measurement could be used to test compatibility with a theoretical model. However, one is often concerned with introducing bias / under-coverage in unfolded measurements, typically via regularisation but in this case also through the choice of prior. One study could be to perform the same chi2 test in the reconstructed space, by forward folding the theory spectrum, to check that a similar agreement is seen in both the reconstructed and unfolded space. Such a test in the Eikos method would show additional robustness.

10. Figure 15 (+caption) runs off the page, please fix the formatting.

---

## Round 3 · Referee Report · Anonymous · 2019-9-19

Strengths

1 - the manuscript presents in a clear manner the topic and provides a fully worked out example of its application
2 - the figures are exhaustive of the results of the application of the method
3 - a link to a fully accessible code implementing the described method is provided

Weaknesses

1 - the reference list is rather poor
2 - a crucial test of the consistency of unfolding (see below) is not performed

Report

This manuscripts presents an adaptation of Bayesian unfolding that suits an important problem in particle physics and nuclear physics applications.
The method is described clearly and the text then proceeds to demonstrate its application to a toy problem.

I found the article in general clearly written, although some more care would be advisable when defining the variables in the offered equations (see detailed changes requested below).

The toy problem is of adequate complexity, but I found that it would have been more interesting to show the performance of the algorithm and its comparison to other methods over a set of different examples, as it is hard to judge on the merits of the proposed method with only one application - the risk that it was voluntarily or involuntarily chosen ad hoc exists.

The author discusses two tests of the method results, but one important test of consistency is not offered. This would consist in providing a proof that the inferential power on generative parameters in the folded and unfolded space are similar, in the presence of a flat prior; i.e. that the unfolding cannot improve the discriminative power of incorrect truth models. In practice, one would compare the discrimination of the unfolded spectrum with respect to different truth models to the discrimination of the folded data with respect to different smeared truth models. The test could be performed with some simple comparison of chisquared values obtained from differences between data and model in the folded and unfolded space.

The captions to the figures are rather short, more detail and explanation of what is shown and what is the point of the figures seems appropriate.

Requested changes

1 - cross-section should not hyphenated unless there is a trailing adjective
2 - Formula 4 has some undefined and non properly indexed variables (eg. x)
3 - In the line following eq. 4, what is X ? Define it.
4 - On page 4, L1: the shifts DS, DB may cause the total counts to go negative, as the gaussians have unlimited support. Please comment in the text how this is addressed.
5 - "An additional K parameters" -> "K additional parameters"
6 - This sentence is unclear: "center of a reasonably lareg .... distribution", please rewrite.
7 - What are "positive-definite Gaussian"s? Maybe "truncated Gaussians"? Please clarify the text.
8 - On page 5 "is equivalent to an unregularized matrix inversion" -> does this not depend on the prior being flat?
9 - Last sentence of page 5 "to favour smooth ... as smoothness" is awkward, please rewrite.
10 - it is not clear why the toy model includes acceptance and efficiency, when these are chosen to be flat. In other words, one cannot see whether the model would correctly handle cases when these factors are varying, making their inclusion not useful.
11 - Fig.3 caption "similated" -> simulated
12 - page 9 line 2 "which results" -> "whose results".
13 - page 10 center "loosing" -> "losing"
14 - caption of fig.9: please explain what are the labeled sources of systematic.
15 - The figure in page 22 overflows the page; the caption is missing.
16 - P.22 line 3 "the Bayes' theorem" -> "Bayes' theorem".

---

## Editorial Decision

editor-in-charge_assigned